# Identification and Characterization of *PRE* Genes in Moso Bamboo (*Phyllostachys edulis*)

**DOI:** 10.3390/ijms24086886

**Published:** 2023-04-07

**Authors:** Sujin Zheng, Kihye Shin, Wenxiong Lin, Wenfei Wang, Xuelian Yang

**Affiliations:** 1College of Life Science, Fujian Agriculture and Forestry University, Fuzhou 350002, China; 2College of Horticulture, Fujian Agriculture and Forestry University, Fuzhou 350002, China; 3Department of Microbiology and Immunology, Jeju National University College of Medicine, Jeju 63243, Republic of Korea

**Keywords:** HLH, PRE, fast-growth, shoot elongation, moso bamboo, transcription factor, flowering

## Abstract

Basic helix–loop–helix (bHLH)/HLH transcription factors are involved in various aspects of the growth and development of plants. Here, we identified four HLH genes, *PePRE1*-*4*, in moso bamboo plants that are homologous to *Arabidopsis PRE* genes. In bamboo seedlings, *PePRE1/3* were found to be highly expressed in the internode and lamina joint by using quantitative RT-PCR analysis. In the elongating internode of bamboo shoots, *PePRE* genes are expressed at higher levels in the basal segment than in the mature top segment. Overexpression of *PePRE*s (*PePRE*s-*OX*) in *Arabidopsis* showed longer petioles and hypocotyls, as well as earlier flowering. *PePRE1* overexpression restored the phenotype due to the deficiency of *AtPRE* genes caused by artificial micro-RNA. *PePRE1-OX* plants showed hypersensitivity to propiconazole treatment compared with the wild type. In addition, PePRE1/3 but not PePRE2/4 proteins accumulated as punctate structures in the cytosol, which was disrupted by the vesicle recycling inhibitor brefeldin A (BFA). *PePRE* genes have a positive function in the internode elongation of moso bamboo shoots, and overexpression of PePREs genes promotes flowering and growth in *Arabidopsis*. Our findings provided new insights about the fast-growing mechanism of bamboo shoots and the application of *PRE* genes from bamboo.

## 1. Introduction

Moso bamboo (*Phyllostachys edulis*) is one of the fastest growing non-timber tree plants all over the world, with considerable ecological, economic, and cultural value [1,2]. In spring, it can grow up to 1 m in less than 24 h and reach a final height of 20 m in 45–60 days [3]. The rapid expansion of bamboo stems is driven by the cell division and elongation of internodes, which are regulated by a combination of endogenous phytohormones and environmental factors such as auxin, gibberellin acid (GA), brassinosteroids (BRs), and light [4]. Recent studies demonstrate that the BR, auxin, GA, and phytochrome pathways converge through direct interactions among their transcription factors/regulators, then pass to a tripartite module of helix-loop-helix (HLH) and basic helix–loop–helix (bHLH) factors, which is named the HHbH module [5]. The HLH/bHLH cascade regulates cell elongation downstream of various hormonal and environmental signaling pathways [6,7].

Paclobutrazol-resistant (PRE) proteins are a typical bHLH transcription factors, homologs to the human Id-1 (inhibitor of DNA binding 1) protein which lacks the basic domain required for DNA binding. PRE proteins dimerize with other bHLH factors to inhibit their DNA-binding activity [8,9]. IBH1 (ILI1 binding bHLH protein) interacts with HBI1 (homolog of *BEE2* interacting with IBH1), a positive regulator of cell elongation, and inhibits its transcriptional activity, thereby promoting the hypocotyl elongation by inhibiting the DNA-binding activity of HBI1 [10,11]. PRE1 and ILI1 promote cell elongation both in *Arabidopsis* and rice by interacting with IBH1 and forming a pair of antagonistic HLH/bHLH transcriptional factors that function downstream of BZR1 (brassinazole resistant 1) to mediate BRs regulation of cell elongation [12]. PIF4 (phytochrome-interacting factor 4) had a major role in the multiple signal integration for plant growth regulation [13]. PIF4 and BZR1 are direct targets of PRE1, PRE5, and PRE6/KIDARI [14]. *PRE3*/*TOM7* is involved in regulating a plant’s growth response to light signals [9]. PAR1 (phytochrome rapidly regulated 1)–PRE1 and PAR1–PIF4 heterodimers form a complex HLH/bHLH network that controls cell elongation and plant development in response to light [15]. PRE6 regulates photomorphogenesis by inhibiting the activity of HFR1 (the long hypocotyl in far-red 1) [16,17]. Another study showed the CIB1 (cryptochrome-interacting bHLH 1)–PAR1 and PIF4–PAR1/HFR1 systems antagonistically regulate cell elongation in response to light and high temperature [18]. Auxin works independently of and in conjunction with the PIF and GA pathways to regulate expression of growth-associated genes in cell elongation [19,20].

On the other hand, GA, BR, and auxin enhance cell elongation by inhibiting many inhibitory bHLH factors by inducing the expression of *PRE* family genes [6,16]. Overexpression of *PRE1* suppressed GA-deficient phenotypes of the *ga2* mutant and impacted several aspects of GA-dependent response, indicating that PRE1 plays a regulatory function in GA-dependent development in *Arabidopsis* [21]. Overexpression of *PRE3* suppresses the dwarf phenotype of the *bri1-301* mutant, indicating *PRE*s are involved and functional redundancy in BR signaling and response [9]. PRE1 acts downstream of ARF10 (auxin response factor 10) in regulating hypocotyl elongation, whereas PRE6 is a transcriptional repressor that is directly regulated by ARF5 and ARF8 in *Arabidopsis* [22].

PRE homologs are also involved in the regulation of plant development in other species. In rice, the *PRE* homologous gene *OsILI1* (*increased leaf inclination 1*) stimulates cell elongation in the lamina joint and *OsILI1* overexpression results in a large leaf angle phenotype [10]. FaPRE (*Fragaria × ananassa* paclobutrazol resistant) promotes the expression of genes involved in the ripening process while suppressing the expression of growth-promoting genes in the receptacle of octoploid strawberries [23]. *GhPRE1* (*Gossypium hirsutum* paclobutrazol resistant 1) overexpression results in longer fibers with higher quality characteristics in cotton [24]. In a recent study, moso bamboo *PRE* homolog genes were downregulated by a BR biosynthesis inhibitor (propiconazole (PPZ)) treatment, according to the transcriptome profile [25].

Recently, transcriptome sequencing and epigenetic-modification profiling in fast-growing moso bamboo shoots identified a large number of putative fast-growing genes [3]. Among these genes, the transcription factors act as regulatory switches for gene expression, which will deepen and expand understanding the fast-growth mechanism. However, the role of moso bamboo *PRE* genes, putative growth-promoting transcription factors, remains unclear. In this study, we identified four *PRE* genes from the moso bamboo genome. Specific expression patterns of *PePRE* genes were observed in the seedling and elongating bamboo shoots, as well as in the different parts of the internodes. Overexpressing *PePRE*1 in *Arabidopsis* resulted in phenotypes including a long petiole, slightly pale green leaves, and early flowering. PRE1 and PRE3 proteins accumulated as punctate structures in the cytosol. Our study suggests that *PePRE*s may play a positive role in the fast-growth processes of moso bamboo. Our results will benefit the future identification of more growth-promoting genes from moso bamboo genome and will provide basis for further functional characterizations of *PRE* family genes.

## 2. Results

### 2.1. Identification of PePREs in Moso Bamboo

Four homologous *PRE* genes were identified from the moso bamboo genome by BLAST-P searching in the moso bamboo protein database (http://www.bamboogdb.org/, accessed on 8 May 2017) [26] with AtPREs amino acids sequences. These were named as *PePRE1* (PH01000065G2010), *PePRE2* (PH01000068G1560), *PePRE3* (PH01000519G0840), and *PePRE4* (PH01000960G0260). When compared with rice and *Arabidopsis* PREs, all bamboo PRE proteins are clustered with rice PREs but not AtPREs (Figure 1b). Protein sequence alignment of the PREs showed that all bamboo PREs proteins have highly conserved helix–loop–helix (HLH) domains but not basic domains that are critical for DNA binding (Figure 1a). The expression patterns of *PePRE* genes were investigated in different tissues of moso bamboo seedlings by quantitative reverse transcription PCR (qRT-PCR) (Figure 1c,d). *PePRE2* and *PePRE3* are found highly expressed in roots, whereas *PePRE1* and *PePRE4* were hardly detected (Figure 1d). Interestingly, only the *PePRE1* gene was detected in leaves, suggesting its unique function in leaf development (Figure 1d). Consistent with a previous report about the critical function of *PRE* genes on the lamina joint bending [12], four bamboo *PRE* genes were found to be expressed in the lamina joint (Figure 1d). As all *PePREs* were expressed in the internode and sheath, *PePRE1* and *PePRE3* showed predominant expressions in the sheath and internode, respectively (Figure 1d). Together, these results suggested that *PePRE* genes showed tissue-specific patterns in moso bamboo seedlings.

### 2.2. Expression of PePRE Genes in Elongating Bamboo Shoots

Bamboo shoot growth is attributed to cell proliferation in the intercalary meristem and subsequent cell elongation in the elongation zone of the internodes. The expression patterns of bamboo *PRE* genes were investigated in the elongating bamboo shoot (Figure 2a). *PePRE1* and *PePRE3* transcript levels were highest in the elongating internode (EIN) and scale leaves, whereas *PePRE2* and *PePRE4* transcript levels were highest in the scale leaves (Figure 2b). All *PePRE*s showed higher expression levels in the internodes than in the nodes (Figure 2b). To further explore the role of *PePRE* genes in internode growth, the elongating internode of bamboo shoots was divided into top, middle, and basal regions (Figure 2a bottom right). Generally, expression of *PePRE* genes accumulated in the middle and basal regions, whereas *PePREs* were hardly detected in the upper parts. *PePRE1* and *PePRE3* were predominantly expressed in the middle and basal regions (Figure 2c). There was high *PePRE2* expression in scale leaves, followed by the unelongated internodes. Interestingly, *PePRE4* showed the lowest expression levels in the internodes and had relatively higher expression in the upper parts. Collectively, these data suggest that *PePRE*s have a prevalent accumulation in the elongating tissue of bamboo shoots.

### 2.3. Overexpressing PePREs in Arabidopsis

To further understand the biological function of *PePREs*, we generated overexpression transgenic lines with the 35S promoter driving the *PePRE* cDNAs fused with green fluorescent protein (GFP) in a Col-0 background. A total of 59, 42, 64, and 14 transgenic lines for PePRE1, 2, 3, and 4 were generated, respectively, with 31 (52.5%), 34 (81%), 22 (34.4%), and 3 (21.4%) lines for each gene, respectively, showing an early flowering phenotype. Days to bolting for *PePRE1-OX* #19 and *PePRE2-OX* #21 were found to be reduced by 13.3% and 14.0%, respectively, whereas the total rosette leaf numbers of *PePRE-OX* plants were also reduced with exception of *PePRE4-OX.* Under long-day conditions, *PePRE1-OX* transgenic plants exhibited longer petioles and had paler green leaves than control plants (Figure 3a), which were similar to the *AtPRE1* overexpression lines reported in a previous study [27]. This suggested that *PePREs* have a conserved function similar to *AtPRE1* on flowering control. A total of 45% and 82.5% of *PePRE1* and *PePRE3* overexpression transgenic lines, respectively, showed a stem-bending phenotype, which was mainly due to a broken stem with a longitudinal crack (Figure A2). We further crossed *PePRE1-OX* transgenic plants with *pre-amiR* transgenic lines, in which four *AtPREs* (*AtPRE1/2/5/6*) were knocked-down using artificial microRNA [14]. Due to the lack of *AtPRE*s, *pre-amiR* exhibited extreme dwarfism, delayed flowering, and had a reduced fertility phenotype [21]. All F1 plants showed a normal phenotype similar to the wild type (Figure 3d). Overexpression of *PePRE1* rescued the *AtPRE* (*1*/*2*/*5*/*6*)-deficient phenotypes of the *pre-amiR* mutant, including late flowering and curled leaves. To exclude *AtPRE1* expression interference, the expression level of *AtPRE1* was detected in all plants. Only in the *AtPRE1-OX* line did the *AtPRE1* expression level increase; there was no difference in the Col-0 and *PePRE1*-*OX* lines (Figure 3e). These studies suggest that *PePRE* genes play a conserved role similar to *AtPRE* genes on flowering promotion and cell elongation.

### 2.4. Subcellular Localization of PePREs in Arabidopsis

To determine the intracellular localization of PePRE proteins, the fluorescence signals of 35S::*PePRE-GFP* were examined in transgenic *Arabidopsis* plants. PePRE-GFP signals were mainly located in the nucleus, cytosol, and plasma membranes, whereas 4′, 6-diamidino-2-phenylindole dihydrochloride (DAPI) staining was used as the indicator of nuclear area. Interestingly, PePRE1-GFP and PePRE3-GFP proteins specifically displayed small punctate structures in the cytosol (Figure 4a), which are different from the AtPRE1 protein. When treated with BFA, a vesicle trafficking inhibitor [28], aggregation of PePRE1 fusion proteins in the punctate structures was reduced. Moreover, these signals were recovered by removing BFA (Figure 4b). Taken together, PePREs are located in the nucleus, cytosol, and membrane, whereas PePRE1 and PePRE3 are distributed in the cytosol in punctate structures.

### 2.5. BR Regulates the Expression Levels of PePREs

The expression levels of *PePRE1* and *PePRE2* dramatically decreased in the aerial part, whereas BR increased the expression of *PePRE1* and *PePRE2* in bamboo shoot [25]. We further examined the expression profile of *PRE* homologous genes in various tissues of bamboo seedlings with PPZ and eBL (Figure 5). Expression of *PRE1*, *3,* and *4* was decreased by PPZ but recovered by subsequent BR treatment in most tissues. However, mRNA levels of *PePRE2* increased in the internode and sheath and decreased in the lamina joint and root. In addition, *PePRE2* did not further respond to eBL treatment. Overexpression of the *PePRE1* gene resulted in longer hypocotyl in comparison with the wild type (Figure 6a,b; Figure A1). Compared with the wild type, *PePRE1* overexpressing plants have longer hypocotyls and roots (Figure 6a,b), similar to *AtPRE1* overexpressing plants. Additionally, compared with *AtPRE1-OX*, the shoot and root parts of *PePRE1-OX* plants showed more sensitivity and less sensitivity to PPZ, respectively (Figure 6c,d; Figure A3).

## 3. Discussion

Although the rapid growth of woody bamboo plants has been widely studied, little is known about the molecular mechanism underlying the elongation of moso bamboo. In this study, we investigated the unique patterns of *PePRE* genes, the conserved growth-promoting function in *Arabidopsis*, and the potential important role for *PePRE*s in the elongation of moso bamboo.

### 3.1. PePREs Have Tissue-Specific Expression Patterns in Moso Bamboo

*PePRE1* and *PePRE4* are highly expressed in the shoot, whereas *PePRE2* and *PePRE4* are specifically expressed in the roots of seedlings. Consistent with these tissue-specific expressions, a rice homologous gene of *PePRE*, *PGL1* (positive regulator of grain length 1), was expressed in the floral organs, young panicle, and predominantly in the root but not the leaf [29]. The expression level of *BU1* (brassinosteroid upregulated 1) is high in the lamina joint in vegetative organs and the panicle at the heading stage [30]. *OsILI1* is ubiquitously expressed in rice, whereas the highest expression of *OsILI1* was observed in the lamina joint [12]. *OsBUL1* (*O. sativa* brassinosteroid upregulated 1-like1), an *AtPRE* homolog in rice, is preferentially expressed in the lamina joint where it controls cell elongation and positively affects leaf angles [31,32]. *FaPRE1* was proposed as a ripening-associated gene and showed a rapid increase in expression in the receptacle during fruit enlargement [33]. In our study, all four *PRE* genes were highly expressed in the elongating tissues but not in the mature tissues such as the nodes of the bamboo shoot (Figure 1 and Figure 2), which is consistent with their function in promoting cell elongation. During the rapid growth of monocots, elongation generally occurs from top to bottom in each individual internode [34]. From our results, *PePRE1* is highly expressed in the basal part of the elongating internodes, with a gradient distribution from the basal to the top parts (Figure 7), which may contribute to the fast growth of bamboo shoot. A previous report showed that *GRF*s (growth-regulating factors) and *ARF* genes were highly expressed in the basal region of the elongating internode, in which *ARF6* and two *ARF8*s targeted by *miR167* and 11 *GRFs* targeted by *miR396* were in the top region [35]. The *DsEXLA2* (*Dendrocalamus sinicus* expansin-like A2) gene is highly expressed in the elongating internode and accelerates the plant growth rate of *Arabidopsis* [36]. *PeGT43*s (*P. edulis* Glycosyltransferase 43) and lignin biosynthesis are significantly upregulated within the shoot [3]. Considering the roles for *PRE* in *Arabidopsis*, *PePRE* genes may work downstream of *ARF* or *GRF* genes to regulate target genes in the elongating bamboo.

### 3.2. Phytohormones Regulates PRE Function in Bamboo Elongation

The stem elongation was contributed to by the division and expansion of individual internodes [37]. Transcriptome analysis revealed that multiple signaling pathways, including GA, auxin, and ABA, may play a role in regulating internode elongation [38]. GA plays an antagonistic regulatory role in regulating internode stem elongation in rice [39]. *OsbHLH073* encodes an atypical bHLH protein and regulates plant height, internode elongation, and panicle extension by regulating GA biosynthesis genes [40]. The content of GAs in dwarf bamboo varieties is also lower than that in normal bamboos, implying that GA plays a major role in the height of Shidu bamboo [41]. Exogenous application of GA resulted in a significant increase in internode length in bamboo seedlings [42]. Those results hint that GA may play a dual role in internode elongation between shoots and seedlings of bamboo. PREs were reported to act as hostile antagonists of the bHLH family of transcription factors, which positively regulate cell elongation via multiple signaling pathways [6,7].

Bamboo PRE proteins showed different subcellular localizations, and only PRE1 and PRE3 showed punctate structures that were disturbed by BFA treatment. BFA treatment may lead to rapid protein aggregation within the endoplasmic reticulum and collapse of the Golgi apparatus [28]. The specific localizations of PePRE1 and PePRE3 may be related to the tuning of protein stability or function. PePREs are localized in both the nucleus and cytoplasm, whereas the punctate structures in cytoplasm were only observed in PePRE1 and PePRE3 in transgenic lines (Figure 4). Considering synchrony in phenotype, expression pattern, and subcellular localization, we proposed that the difference in subcellular localization may contribute to its functional variation. Certain *PePRE* overexpressing plants were easily dislodged caused by a broken stem (Figure A2). PRE1 was reported to promote cell elongation by preventing IBH1 from inhibiting HBI1, which directly activates genes encoding cell-wall-loosening enzymes (e.g., EXP, etc.) [10,43]. The accumulation of PePRE may change the stem segment structure, especially the composition of the cell wall, leading to the rupture and hollowness of the stem (Figure A2). 

### 3.3. Various Functions of PRE in Regulating Plant Growth through Multiple Signaling Pathways

PRE proteins are important parts of the signaling pathways involved in physiological development and reproduction [21,27,44]. Due to the limitation of the transformation of moso bamboo, the function of *PePRE*s was checked in *Arabidopsis* in this work. The overexpression of *PePREs* was associated with early flowering and promoted growth, such as longer petioles, hypocotyls, and roots (Figure 7). *PREs* are involved in regulating the growth of floral organs in *Arabidopsis* [27,45]. *FaPRE1* antagonistically modulates the transcription of genes related to both receptacle growth and ripening [23,33]. *GhPRE1* has contributed to spinnable fiber formation in cotton; overexpressing *GhPRE1* leads to longer fibers with improved quality parameters, indicating that this bHLH gene is useful for improving cotton fiber quality [24]. *SlPRE2* was reported to regulate fruit development via the gibberellin pathway and tomato fruit pigment accumulation in tomatoes [46]. Those results suggest that *PRE* affects multiple aspects of the development of plants. PRE6 is a positive regulator of shade avoidance and interacts with a number of negative growth regulators (PAR1, etc.) [17]. Transcriptional regulators (ARFs and BZR1) and post-transcriptional regulators (HFR1, etc.) were key modules of the signaling network controlling shade avoidance [47,48]. Interestingly, *PePRE* genes are regulated by brassinosteroid levels (Figure 5) in the elongating bamboo, and the transgenic lines also shown an altered response to PPZ treatment (Figure 6). Shade avoidance syndrome (SAS) allows plants that are grown in densely populated environments to maximize their sunlight access [48]. As mentioned above in the relationship between those TFs and PRE, PePREs may play a positive role in rhizome elongation underground and shoot elongation aboveground. Functional analyses of PePREs will help to elucidate the mechanism of fast growth in plants as well. Light affects the dynamic growth and development of the *P. pygmaeus* rhizome–root system [49]. Taken together, *PePRE* genes show a conserved function in controlling flowering and promoting growth by responding to multiple signaling pathways. This is similar to the *PRE* genes from other species. 

Our analysis identified *PePRE* genes with specific expression patterns in the seedling and shooting stage and demonstrated that *PePREs* are brassinosteroid-regulated genes. Overexpression of the *PePRE1* gene will promote flowering, hypocotyl elongation, and root growth. The current results indicate a key role for *PePRE* genes in bamboo growth and development. Our findings will provide the basis for further functional characterizations of *PRE* family genes and the molecular mechanism of bamboo fast growth and will benefit the application of growth-promoting gene resources from bamboo. 

## 4. Materials and Methods

### 4.1. Plant Material and Growth Conditions

The moso bamboo shoots (approximately 1.8 m above ground height) were obtained from the Bamboo Garden of Fujian Agriculture and Forestry University, Fuzhou (coordinate 119°14′ E, 26°50′ N). Moso bamboo seeds were collected from Gong city, Guanxi province, China (118°48′ E, 24°51′ N). The seeds of moso bamboo were soaked in tap water for 24 h to induce seed germination and then sown on the soil. Seedlings were cultivated for 3 weeks in a greenhouse (long-day conditions, 22 °C).

Columbia (Col-0) wild-type seeds of *Arabidopsis* were germinated on half MS medium (pH = 5.8) with 1% sucrose, then transferred to a greenhouse (long-day conditions, 22 °C). *PePRE-OX* was crossed with *pre-amiR* to generate *PePRE-OX pre-amiR* plants. 

### 4.2. Protein Sequence Alignment and Phylogenic Tree Construction

Alignment of the protein sequences was performed using Clustal Omega and analyzed in the GENEDOC program with the default settings. A phylogenic tree based on the sequence alignment was generated using MEGA-X by the neighbor-joining method [50]. Accession number: The protein sequences reported in this article can be found in the database (http://www.bamboogdb.org/, accessed on 8 May 2017) [26], the rice genome database (http://rice.plantbiology.msu.edu/, accessed on 8 May 2017), and TAIR (http://www.arabidopsis.org/, accessed on 8 May 2017) under the following accession numbers: *OsILI1* (Os04g54900.1), *OsILI2* (Os11g39000.1), *OsILI3* (Os03g07540.1), *OsILI4* (Os06g12210.1), *OsILI5* (Os02g51320.1), *OsILI6* (Os03g07510.1), *OsILI7* (Os10g26460.1), *PePRE1* (PH01000065G2010), *PePRE2* PH01000068G1560), *PePRE3* (PH01000519G0840), *PePRE4* (PH01000960G0260), *AtPRE1* (AT5G39860.1), *AtPRE2* (AT5G15160.1), *AtPRE3* (AT1G74500.1), *AtPRE4* (AT3G47710.1), *AtPRE5* (AT3G28857.1), and *AtPRE6* (AT1G26945.1).

### 4.3. Gene Expression Analysis

Total RNA was isolated from various tissues with the Plant Total RNA Kit (Sigma, ATRN50) and extensively treated with RNase-free DNase I (Sigma, St. Louis, MO, USA, DNASE70-1SET). cDNA was generated by RT-PCR using the PrimeScript^TM^ RT reagent Kit (Takara, Kusatsu, Japan, RR047A). qRT-PCR analysis was performed with SYBR Premix Ex Taq II (Takara, RR820) on a QuanStudio 6 Flex instrument (Applied Biosystems, Waltham, MA, USA). For each sample, qPCR was performed with three technical replicates on three biological replicates. The bamboo *PeTIP41* gene [51] was used as an internal control for the qRT-PCR. The relative expression levels were calculated as E^−ΔCq^ and normalized to *PeTIP41*. The primer sequences used in this study are listed in Table A1.

### 4.4. Vector Construction and Transformation

The full-length *PePRE1* sequence was amplified and ligated into pAGM1311. A 35S-promotor-driven PePRE C-terminal fused with a GFP tag was constructed into a pAGM4673 backbone. Through *Agrobacterium tumefaciens s*train GV3101, these constructions were transformed into *Arabidopsis* (Col-0). T1 seeds were screened by an RFP selection marker and then further confirmed by qRT-PCR and Western blot. 

### 4.5. Hypocotyl Measurements and Statistical Analysis

For hypocotyl and root length measurements, seedlings were grown for 7 days on vertically oriented plates. Seedlings were flattened and photographed before taking quantitative measurements using ImageJ software (http://rsb.info.nih.gov/, accessed on 8 April 2018) to analyze the scanned images of the seedlings. The differences among groups were assessed by one-way ANOVA and Duncan’s multiple comparisons test using SPSS 23.0. GraphPad Prism 8.0 (http://www.graphpad.com/, accessed on 26 April 2018) was used to plot figures. At least 20 seedlings were measured, and experiments were repeated more than two times.

### 4.6. BFA Treatment

For BFA treatment, 4-day-old seedlings of *PePRE1-OX* were incubated in 50 μM BFA for 90 min before viewing the seedlings; control seedlings were incubated in a solution without BFA but with DMSO at the same concentration as the BFA-treated seedlings. After BFA treatment, the seedlings were washed with half MS liquid media several times and the root tips were observed for the formation of BFA compartments using a confocal laser scanning microscope (Leica Microsystems, Wetzlar, Germany). For recovery, BFA was removed from seedlings and supplied with half MS media for another 40 min followed by immediate observation with the confocal microscope.

## 5. Conclusions

Our study identified an atypical bHLH transcription factor (*PRE* homologs) in moso bamboo via gene expression analysis, heterologous overexpression, etc. We verified that *PePREs* function as a positive regulator in the promotion of internode elongation during the fast-growth process. Overexpressing *PePRE* promoted *Arabidopsis* growth and petiole/hypocotyl elongation. Our findings shed light on bHLH-mediated fast growth to provide preliminary knowledge for fast-growing plants.

## Figures and Tables

**Figure 1 ijms-24-06886-f001:**
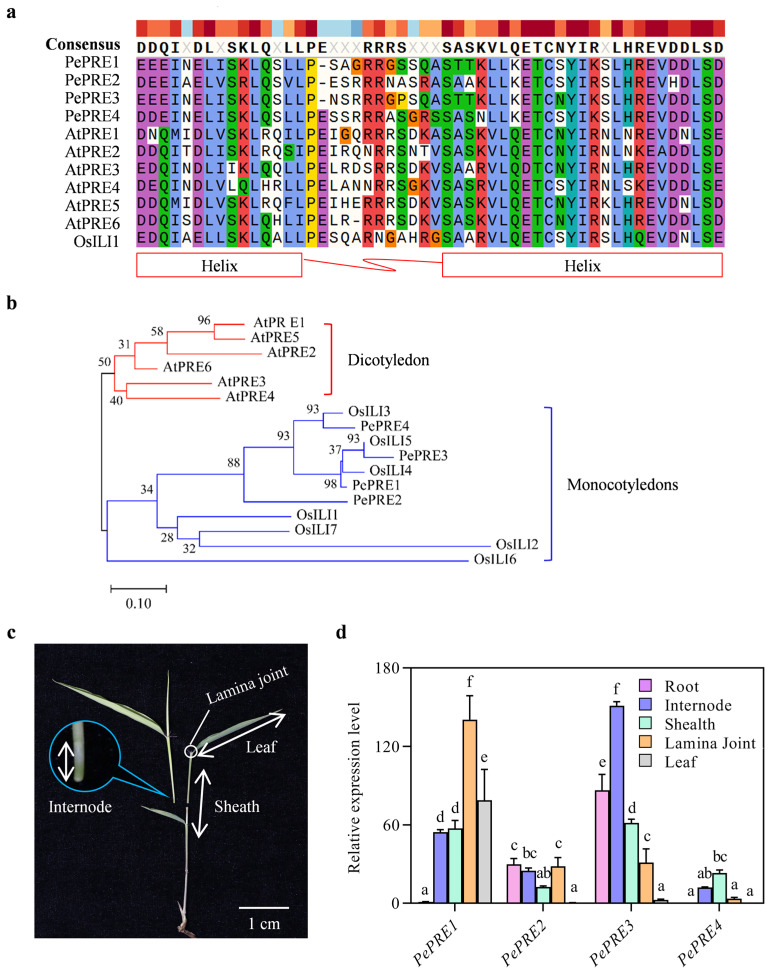
*PePRE*s in moso bamboo. (**a**) Sequences alignment of PRE proteins. Identical or similar amino acid residues are indicated by colorful shading and the helix and loop domains are highlighted. (**b**) Phylogenetic analysis of PRE proteins from rice and *Arabidopsis*. Full amino acid sequences were used. Bootstrap values of 1000 replications were shown. Pe, *Phyllostachys edulis*; *Os*, *Oryza sativa*; *At*, *Arabidopsis thaliana*. (**c**) The tissue of the aerial part of moso bamboo seedling. Scale bar represents 1 cm. (**d**) Expression level of *PePRE*s in different tissues of 3-week-old seedlings by qRT-PCR analysis. Different letters above the data columns indicate significant differences (*p* < 0.05; Duncan’s test).

**Figure 2 ijms-24-06886-f002:**
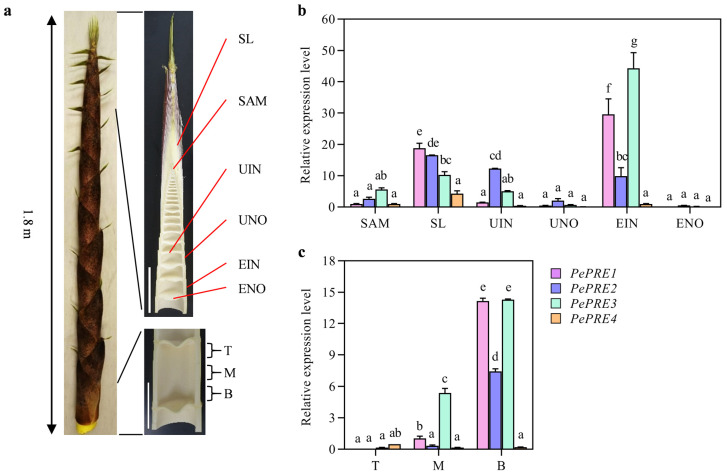
Expression pattern of *PePRE*s in the elongating bamboo shoot. (**a**) Structure of an elongating moso bamboo shoot. Samples are collected from a 1.8 m bamboo shoot. SAM, shoot apical meristem; SL, scale leaf; UIN, unelongated internode; UNO, unelongated node; EIN, elongating internode; ENO, elongating node. Scale bar represents 10 cm. (**b**) Expression levels of *PePRE* genes in the elongating bamboo shoot. (**c**) Expression levels of *PePRE* genes in different parts of an elongating internode. T, top part; M, middle part; B, bottom part. Different letters above the data columns indicate significant differences (*p* < 0.05; Duncan’s test).

**Figure 3 ijms-24-06886-f003:**
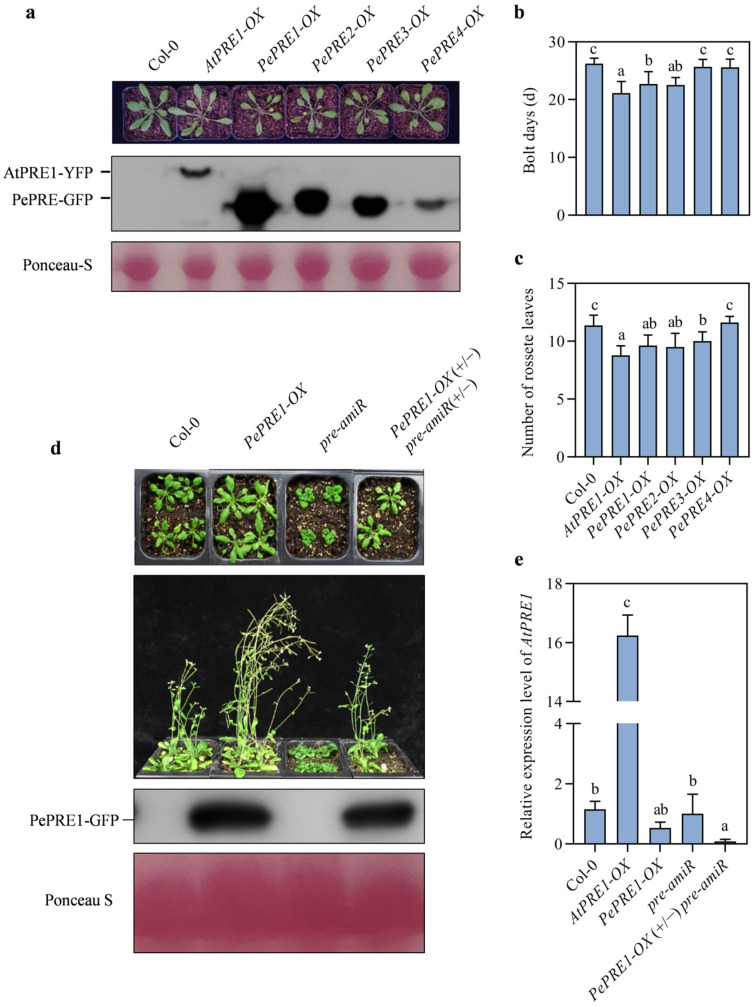
Overexpression of *PePRE*s promoted plant flowering and growth in *Arabidopsis.* (**a**) Overexpression of *PePRE*s in *Arabidopsis* increased petiole elongation. Panels from top to bottom are pictures of plants grown in soil for 4 weeks (left to right: Col-0, *AtPRE1-OX* (35S::*AtPRE1-YFP*), and *PePREs-OX* (35S::*PePREs-GFP*)), Western bolt of PRE proteins using GFP antibody, and protein loading control by Ponceau S. Days of bolting (**b**) and rosette leaf numbers (**c**) when plants started bolting. A total of 10 plants were used for calculations. (**d**) Overexpression of *PePRE1* suppressed the dwarf phenotype of *pre-amiR*. Panels from top to bottom are morphology of 4-week-old plants of Col-0, *PePRE1-OX*, *pre-amiR*, and *PePRE1-OX* (+/−) *pre-amiR* (+/−), 5-weeks-old plants of Col-0, *PePRE1-OX*, *pre-amiR*, and *PePRE1-OX* (+/−) *pre-amiR* (+/−), expression level of PRE1 using GFP antibody, and the protein loading control by Ponceau S. (**e**) Quantitative analysis of *AtPRE1* in Col-0, *PePRE1-OX*, *pre-amiR*, and *PePRE1-OX* (+/−) *pre-amiR* (+/−). Different letters above the data columns indicate significant differences compared between Col-0 and transgenic lines (*p* < 0.05; Duncan’s test).

**Figure 4 ijms-24-06886-f004:**
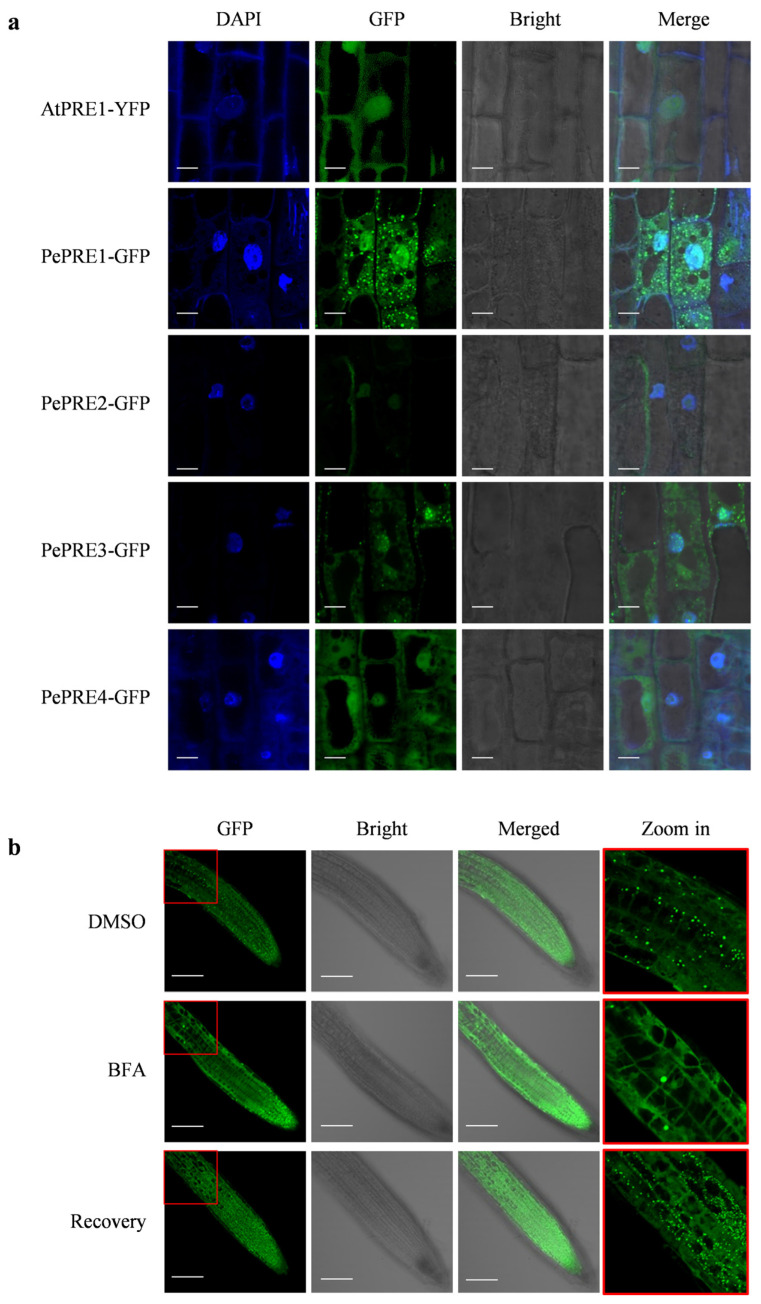
Subcellular localization of PRE proteins in *Arabidopsis*. (**a**) The root tip of the *PePRE*s’ transgenic lines was observed by confocal microscope. Seedlings were grown on half MS medium with constant light for 7 days. Cells were stained with DAPI. Scale bar indicates 10 μm. (**b**) Subcellular localization of PePRE1-GFP by treating with or without BFA. The root tips of *PePRE1*-OX #19 were observed by confocal microscope. Seedlings were grown on half MS medium with constant light for 4 days vertically. Scale bar indicates 100 μm.

**Figure 5 ijms-24-06886-f005:**
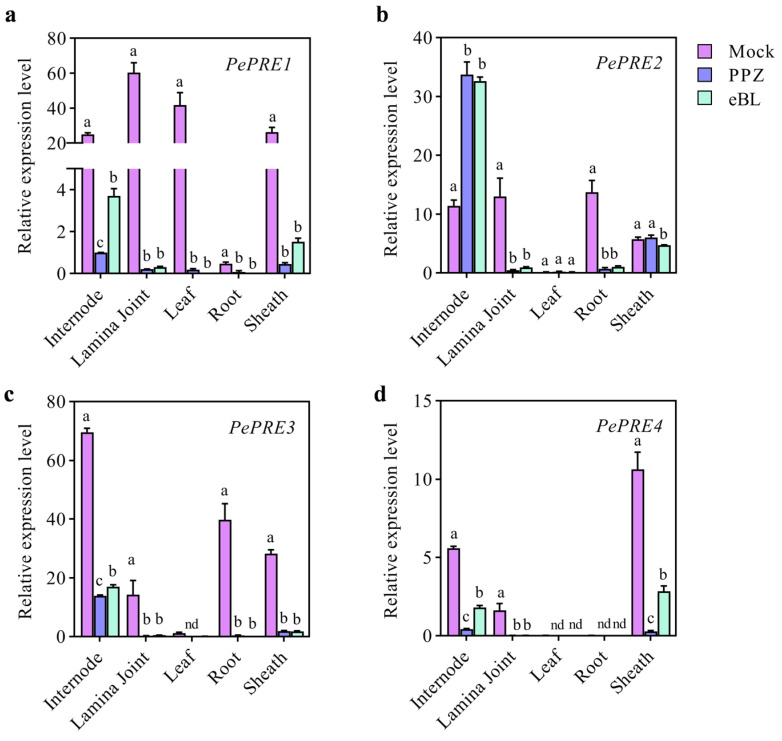
The expression levels of *PePRE*s after eBL or PPZ treatment in bamboo seedlings. Quantitative RT-PCR analysis of *PePRE*s (**a**–**d**) in different bamboo seedling organs after PPZ or eBL treatment. The bamboo seedlings were grown on half MS medium for 18 days. For PPZ treatment, plants were transferred to half MS medium containing 50 μM PPZ for another 4 days. For eBL treatment, the seedlings were transferred to half MS medium containing 10 μM PPZ and 1 μM eBL solution for 4 h. Different lowercase letters above the data columns indicate significant differences (*p* < 0.05; Duncan’s test); nd, no data available.

**Figure 6 ijms-24-06886-f006:**
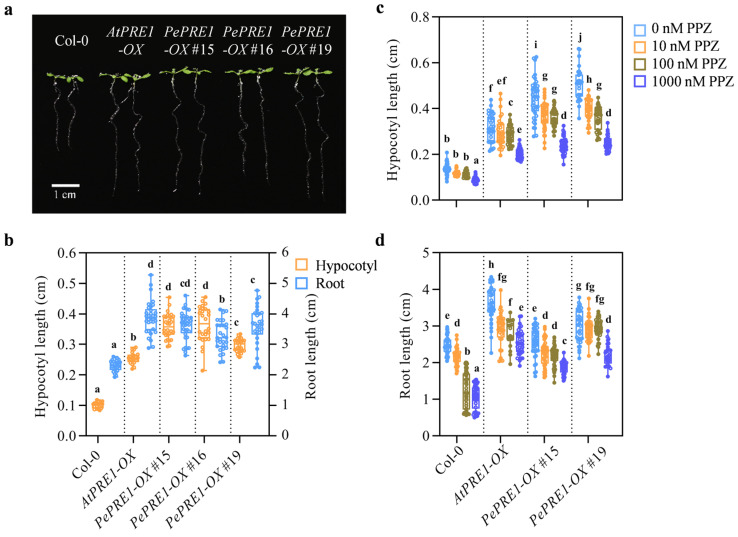
*PePRE1-OX* transgenic *Arabidopsis* exhibited longer hypocotyls and roots that were hypersensitive to PPZ treatment. (**a**) Representative seedlings of Col-0, *AtPRE1-OX*, and *PePRE1-OX*. The seeds were sown on half MS medium and grown for 7 days vertically. Scale bar presents 1 cm. (**b**) The hypocotyl and root lengths of the control and *PePRE1-OX* plants. Hypocotyl length (**c**) and root length (**d**) of different lines of *PePRE1-OX* compared with Col-0 with a gradient concentration (0 nM, 10 nM, 100 nM, and 1000 nM) of PPZ treatment. Values are the means calculated from at least 20 seedlings. Error bars represent mix/max values. Different letters above the data column indicate significant differences (*p* < 0.05; Duncan’s test).

**Figure 7 ijms-24-06886-f007:**
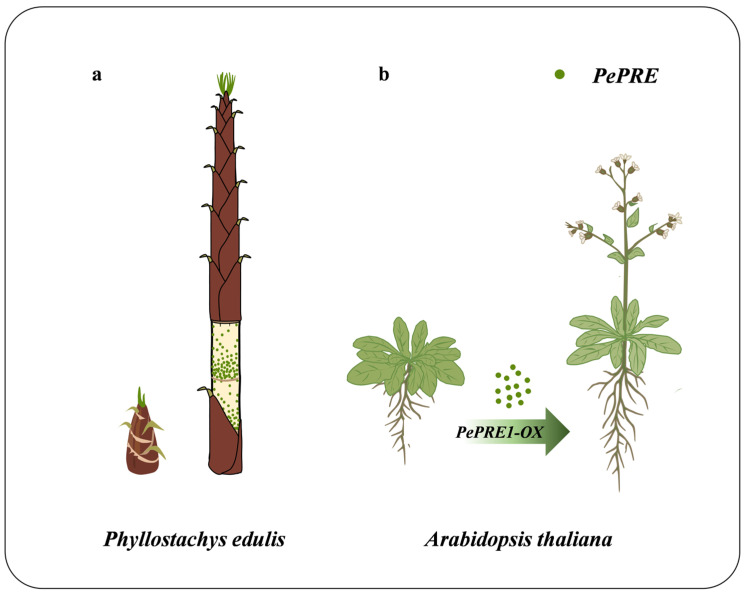
Diagram of *PePRE1* gene expression and function. (**a**) Gradient expression levels of *PePRE1* in the elongating internodes of moso bamboo shoots. (**b**) Overexpression of *PePRE1* genes leads to early flowering, longer hypocotyl, and growth promotion in *Arabidopsis*. Green dots indicate *PePRE* genes.

## Data Availability

Not applicable.

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
