# Peer review of "Identification and Characterization of PRE Genes in Moso Bamboo (Phyllostachys edulis)"

_ijms, 2023, doi:10.3390/ijms24086886_

Round 1

Reviewer 1 Report

Dear Authors,

Reviewer comments ijms-2272920

The manuscript entitled „Identification and characterization of PREs homologous genes in moso bamboo (Phyllostachys edulis)“ represents a useful study aimed at an investigation of paclobutrazol resistant (PRE) proteins and their function in a heterologous system, Arabidopsis thaliana plants overexpressing PePRE1 gene (PePRE1-OX), was determined.

I think that the manuscript presents original work providing novel results which are worth publishing.

However, I have several comments on the present manuscript which are given below:

Results:

In Figure 1b, bootstrap values which are mentioned in the figure legend have to be added to the phylogenetic tree. In Figure 1c, I miss the expression „scale leaf“ used by the authors in the text. In addition, an appropriate scale bar has to be added to the figure. In Figure 1d, statistical evaluation of the significant differences between the individual values indicated by the different letters above the data columns has to be added.

In Figure 2a, an appropriate scale bar has to be added to the scheme of elongating moso bamboo shoot. In Figure 2b and c,  statistical evaluation of the significant differences between the individual values indicated by the different letters above the data columns has to be added.

In Figure 3b,c, e legend, the kind of statistical test used for the determination of significant differences has to specified, i.e., it should be written in the figure legend what do the asterisks in Figure 3b, c and what do the different letters above the data columns in Figure 3e mean for?? In Figure 4 providing the photos of the subcellular localitation of PREs protein in PePRE1-OX Arabidopsis plants, appropriate scale bars have to be added.

Results, line 196: Modify the word form „Compare with“ to „Compared with wild type,…“

Figure 5 legend, line 204: Add the verb „were“ and modify the word order in the statement „For eBL treatment, the bamboo seedlings were transferred into 10 µM PPZ….“

In Figure 5 legend, a short explanation of the kind of statistical test used for the determination of significant differences as indicated by the different letters has to be added.

In Figure 6a, an appropriate scale bar has to be added to the photos of the transgenic Arabidopsis plants. In Figure 6c, the concentrations given in the figure legends have to be specified, i.e., 0 nM, 10 nM, 100 nM, and 1000 nM of what compound??

In Figure 6 legend, line 211? Use the plural form in the statement „Values were the means calculated from at least 20 seedlings.“

Discussion, part 3.1., line 221: Add the verb „are“ preceding the verb „expressed“ in the statement „PePRE1 and PePRE4 are highly expressed in the shoot….“

Discussion, line 247: Use a plural formo f the words „signaling pathways“ in the statement ů…that multiple signaling pathways….“

Line 249: Add the verb „was“ in the statement „it was reported that multiple hormones involved in regulating internode elongation….“

Line 250: Use rather the term „the length“ instead of „the height“ in the statement „GA treatment increased the length of individual internodes….“

Line 264: Add the verb „be“ in the verb „may be related“ in the statement „These specific localizations of PePRE1 and PePRE3 may be related to the tuning of protein stability or function.

Line 267: Add the verb „are“ in the statement „PePREs are localized in both nukleus and cytoplasm….“

Discussion, line 291: I think that the statement should be corrected as follows: „….a positive role in root elongation underground and shoot elongation above the ground.“ I think that it should be written „root elongation underground“ (not „stem elongation underground“) since roots grow under the soil but stems as parts of shoot grow above the soil surface, i.e., aboveground.

Materials and methods, line 302: Modify the verb form „to induced“ to „to induce“ in the statement „The seeds of moso bamboo were soaked in the tap water for 24 hours to induce seed germination,…“

Materials and methods, line 308: Use a singular form of the verb „was performed“ instead of „were performed“ in the statement „Alignment of the protein sequences was performed using Clustal Omega…“

In Discussion, I could recommend the authors to add a summarising figure or a table providing an overview on the biological functions of PRE gene products (proteins) in plants based on their original results based on the PePRE1-OX overexpression in Arabidopsis.

Author Response

Response to Reviewer 1

The manuscript entitled “Identification and characterization of PREs homologous genes in moso bamboo (Phyllostachys edulis)”represents a useful study aimed at an investigation of paclobutrazol resistant (PRE) proteins and their function in a heterologous system, Arabidopsis thaliana plants overexpressing PePRE1 gene (PePRE1-OX), was determined.

I think that the manuscript presents original work providing novel results which are worth publishing.

Reply: Thank you.

Question 1.

In Figure 1b, bootstrap values which are mentioned in the figure legend have to be added to the phylogenetic tree. In Figure 1c, I miss the expression “scale leaf” used by the authors in the text. I addition, an appropriate scale bar has to be added to the figure. In Figure 1d, statistical evaluation of the significant differences between the individual values indicated by the different letters above the data columns has to be added.

Reply: Thank you for your suggestion.

We added bootstrap values to the phylogenetic tree (Figure 1b) and statistical evaluation was add to Figure 1d.

For the “scale leaf”, it is a term for the specifical leaf in the bamboo shoot stage (Figure 2 a), and term has also been used by other researchers (such as Upadhyaya et al, 2004, J. Bamboo and Rattan). In Figure 1, we did not mention the scale leaf, since this section is about the expression levels in seedling tissues.

Question 2.

In Figure 2a, an appropriate scale bar has to be added to the scheme of elongating moso bamboo shoot. In Figure 2b and c, statistical evaluation of the significant differences between the individual values indicated by the different letters above the data columns has to be added.

Reply: Thank you. We have added scale bars and statistical evaluation to Figure 2 following your suggestion.

Question 3.

In Figure 3b, c, e legend, the kind of statistical test used for the determination of significant differences has to specified, i.e., it should be written in the figure legend what do the asterisks in Figure 3b, c and what do the different letters above the data columns in Figure 3e mean for??

Reply: Thank you. We have analyzed the data in Figure 3 with Duncan’s test and edited the associated legends.

Question 4.

In Figure 4 providing the photos of the subcellular localization of PREs protein in PePRE1-OX Arabidopsis plants, appropriate scale bars have to be added.

Reply: Thank you. We added scale bars following your suggestion.

Question 5.

Results, line 196: Modify the word form “Compare with” to “Compared with wild type,…”

Reply: Thank you. We modified it as suggestion.

Question 6.

Figure 5 legend, line 204: Add the verb “were” and modify the word order in the statemen. For eBL treatment, the bamboo seedlings were transferred into 10µM PPZ….”

Reply: Thank you. This legend has been corrected.

Question 7.

In Figure 5 legend, a short explanation of the kind of statistical test used for the determination of significant differences as indicated by the different letters has to be added.

Reply: Thank you. We added explanation of statistical test in Figure 5 legend.

Question 8.

In Figure 6a, an appropriate scale bar has to be added to the photos of the transgenic Arabidopsis plants. In Figure 6c, the concentrations given in the figure legends have to be specified, i.e., 0 nM, 10 nM, 100 nM, and 1000 nM of what compound??

Reply: Thank you. We have added scale bar in Figure 6a. In Figure 6c, we also specified the PPZ concentration in legend following your suggestion.

Question 9.

In Figure 6 legend, line 211? Use the plural form in the statement “Values were the means calculated from at least 20 seedlings.”

Reply: Thank you. We changed “Value” to its plural form “Values”.

Question 10.

Discussion, part 3.1., line 221: Add the verb “are” preceding the verb “expressed” in the statement “PePRE1 and PePRE4 are highly expressed in the shoot….”

Reply: Thank you. We corrected it.

Question 11.

Discussion, line 247: Use a plural form of the words “signaling pathways” in the statement…that multiple signaling pathways….”

Reply: Thank you. We corrected it as your suggestion.

Question 12.

Line 249: Add the verb “was” in the statement “it was reported that multiple hormones involved in regulating internode elongation….”

Reply: Thank you. We added “was” in the right position in the text.

Question 13.

Line 250: Use rather the term “the length” instead of “the height” in the statement “GA treatment increased the length of individual internodes….”

Reply: Thank you. We changed “the height” to “the length” in the text.

Question 14.

Line 264: Add the verb “be” in the verb “may be related” in the statement “These specific localizations of PePRE1 and PePRE3 may be related to the tuning of protein stability or function.”

Reply: Thank you. We edited it as suggestion.

Question 15.

Line 267: Add the verb “are” in the statement “PePREs are localized in both nukleus and cytoplasm….”

Reply: Thank you. We edited it as suggestion.

Question 16.

Materials and methods, line 302: Modify the verb form “to induced” to “to induce” in the statement “The seeds of moso bamboo were soaked in the tap water for 24 hours to induce seed germination,…”

Reply:

Reply: Thank you. We corrected it.

Question 17.

Materials and methods, line 308: Use a singular form of the verb “was performed” instead of “were performed” in the statement “Alignment of the protein sequences was performed using Clustal Omega…”

Reply: Thank you. We corrected it as suggestion.

Question 18.

Discussion, line 291: I think that the statement should be corrected as follows: “…. a positive role in root elongation underground and shoot elongation above the ground.” I think that it should be written “root elongation underground “(not “stem elongation underground”) since roots grow under the soil but stems as parts of shoot grow above the soil surface, i.e., aboveground.

Reply: Thank you. We modified it to “a positive role in rhizome elongation underground and shoot elongation aboveground.”

Question 19.

In Discussion, I could recommend the authors to add a summarising figure or a table providing an overview on the biological functions of PRE gene products (proteins) in plants based on their original results based on the PePRE1-OX overexpression in Arabidopsis.

Reply: Thank you. We add a summarizing Figure 7 following your suggestion. In Figure 7, we summarized the unique expression pattern of PRE gene in the elongating internodes, and the promoting effects of PePRE1-overexpression in Arabidopsis.

Reviewer 2 Report

Reviewers' comments:

Manuscript Number: ijms-2272920

Title: Identification and Characterization of PREs Homologous Genes in Moso Bamboo (Phyllostachys edulis).

Comments: 

This work concerns the “Identification and Characterization of PREs Homologous Genes in Moso Bamboo (Phyllostachys edulis)”. Important points are missing and there are some points that should be revised or corrected. Some important points are mentioned hereafter.

- In the Abstract: the authors need to improve with more specific short results and conclusions, i.e. academic novelty or technical advantages.

- Keywords: add more keywords.

- The introduction section should be improved; more related papers must be discussed and superiority, novelty, critical improvement in this study must be clarified.

- 2.2. Expression of PePRE genes in elongating bamboo shoot – Should be improve.

- 2.4. Subcellular localization of PePREs in Arabidopsis – Should be improve.

- 4.3. Gene expression analysis – Should be improve.

- 4.5. Hypocotyl measurements and statistical analysis – Should be improve.

- 4.6. BFA treatment – Should be improve.

- Conclusion section should be improved.

- Make all references in same format for volume number, page numbers and journal name, because it is difficult to searching and reading.

Based on these, I advise the authors to rectify the above mentioned errors and we hope to re-evaluate the revised manuscript.

Round 2

Reviewer 2 Report

Accept